# Boosting the Performance of Generic Deep Neural Network Frameworks with Log-supermodular CRFs

**Hao Xiong**
Department of Computer Science
The University of Texas at Dallas
Richardson, TX 75083
hao.xiong@utdallas.edu

**Yangxiao Lu**
Department of Computer Science
The University of Texas at Dallas
Richardson, TX 75083
yangxiao.lu@utdallas.edu

**Nicholas Ruozzi**
Department of Computer Science
The University of Texas at Dallas
Richardson, TX 75083
nicholas.ruozzi@utdallas.edu

## Abstract

Historically, conditional random fields (CRFs) were popular tools in a variety of application areas from computer vision to natural language processing, but due to their higher computational cost and weaker practical performance, they have, in many situations, fallen out of favor and been replaced by end-to-end deep neural network (DNN) solutions. More recently, combined DNN-CRF approaches have been considered, but their speed and practical performance still falls short of the best performing pure DNN solutions. In this work, we present a generic combined approach in which a log-supermodular CRF acts as a regularizer to encourage similarity between outputs in a structured prediction task. We show that this combined approach is widely applicable, practical (it incurs only a moderate overhead on top of the base DNN solution) and, in some cases, it can rival carefully engineered pure DNN solutions for the same structured prediction task.

## 1  Introduction

A wide range of machine learning applications can be cast as structured prediction problems, joint prediction problems in which the aim is to produce a vector of correlated outputs rather than a single class label, e.g., semantic image segmentation, stereo matching, etc. Historically, conditional random fields (CRFs) were popular for this class of problems as they (1) allow modeling of the underlying problem structure, (2) can incorporate domain knowledge, and (3) admit efficient (approximate) inference in certain settings. More recently, however, increases in data set and model sizes have made CRF-based approaches less practical. Instead, deep neural networks (DNNs) that can be trained at scale by taking advantage of accelerators like GPUs and TPUs have become increasingly popular.

While deep neural networks yield powerful predictive models, they can be sample inefficient [26] and may fail to produce solutions that respect natural constraints required by the structured output, e.g., if the output is a combinatorial object. In practice, the structured outputs of DNNs can lack simple *smoothness* properties that one would expect in a given domain: (1) in stereo matching, neighboring pixels likely have similar depths, (2) in optical flow estimation, neighboring pixels likely have similar flows, (3) in activity recognition in video, objects/activities are likely to persist between frames, (4) in motion/trajectory prediction tasks, objects in motion tend to stay in motion in the same direction, (5) in image colorization, neighboring pixels likely have similar colors, (6) in segmentation, neighboring

36th Conference on Neural Information Processing Systems (NeurIPS 2022).

pixels are more likely to be part of the same class, and (7) in certain statistical physics applications, neighboring particles tend to behave similarly. As a result, the ability to incorporate this type of soft knowledge could improve the training of DNN models and smooth the structured output.

Motivated by existing approaches that combine DNNs with CRFs, such as structured variational autoencoders [22] and models for stereo depth estimation [24, 31], we propose a generic hybrid model, which takes an existing DNN, combines it with a CRF, and then jointly trains the combined model, for the above smoothing task. In order to accomplish the smoothing, our CRFs will employ log-supermodular potentials. This class of potential functions belongs to the family of so-called *attractive* graphical models [54] as the potential functions encourage agreement between the variables over which they operate. As such, we dub our approach *attractive smoothing*. Our aim is to show that (1) the training and prediction overhead of such a hybrid model can be kept low and (2) a sufficiently expressive CRF can act like a regularizer for the DNNs by smoothing the outputs as described above. Our approach applies in both discrete and continuous settings and, to the best of our knowledge, our approach represents a novel, practical application of continuous log-supermodular CRFs.

Via a series of experiments, we show that attractive smoothing can be applied to yield both qualitative and quantitative performance improvements with only a modest overhead compared to the original DNN in a range of applications (stereo matching, semantic segmentation, and image colorization). Our experiments show that even a simple CRF can smooth local neighborhoods and encourage local consistency. In our experiments, the hybrid approach always resulted in a performance improvement, sometimes dramatically, over the pure DNN architecture. In particular, we observed that even comparatively older architectures can be competitive against (or outperform) more recent architectures simply with the addition of an attractive CRF.

## 2 Preliminaries

We will use CRFs to represent a joint probability distribution over a structured output $y$ given data observations $x$ that factorizes as a product of nonnegative potential functions over a given graph structure. Specifically, given a graph $G = (V, E)$ and $\phi_i(\cdot|x) : \mathcal{Y} \to \mathbb{R}_{\geq 0}$ and $\phi_{ij}(\cdot, \cdot|x) : \mathcal{Y} \times \mathcal{Y} \to \mathbb{R}_{\geq 0}$, a pairwise CRF models $p(y|x)$ as

$$p(y|x) = \frac{1}{Z(x)} \prod_{i \in V} \phi_i(y_i|x) \prod_{(i,j) \in E} \phi_{ij}(y_i, y_j|x),$$

where $Z(x)$ is the normalizing constant that ensures that $p(\cdot|x)$ is a probability distribution.

The graph structure and potential functions are problem dependent – they should be chosen to reflect the structure, correlations, and constraints of the output. In particular, we will be interested in log-supermodular (LSM) potential functions as they encourage agreement between the variables upon which they operate. Formally, a function $f : D \to \mathbb{R}_{\geq 0}$ for $D \subseteq \mathbb{R}^n$ is log-supermodular if $f(x)f(y) \leq f(x \wedge y)f(x \vee y)$, for all $x, y \in D$, where for all $i \in \{1, \ldots, n\}$, $(x \wedge y)_i = \min(x_i, y_i)$ and $(x \vee y)_i = \max(x_i, y_i)$. Note that $D$ should be closed under these operations in order for the above definition to make sense, e.g., $D = \{0, 1\}^n$ or $D = [0, 1]^n$. If the inequality is reversed $f$ is said to be log-submodular (submodular and supermodular functions satisfy a similar inequality with products replaced by sums). A twice continuously differentiable function $f : D \to \mathbb{R}_{\geq 0}$ is log-supermodular on $D$ if and only if for all $i \neq j \in \{1, \ldots, n\}$ and $\overline{x} \in D$, $\frac{\partial^2 \log(f)}{\partial x_i \partial x_j}|_{\overline{x} \in D} \geq 0$, i.e., the off-diagonal elements of the Hessian matrix of $\log(f(\cdot))$ are nonnegative over the domain [51]. From the definition, products of log-supermodular functions are log-supermodular. So, if all of the potential functions are log-supermodular, so is $p(y|X)$.

Log-supermodular (log-submodular) functions arise in a variety of applications and represent classes of functions for which global maxima (minima) can be found efficiently in both discrete and continuous settings [14, 47, 21, 1]. Discrete log-supermodular/log-submodular functions also have a rich history in the study of CRFs in both theory and practice, e.g., [19, 5, 25]. Continuous log-supermodular CRFs enjoy similar theoretical properties to their discrete counterparts [41, 42, 2]. However, despite the wide use of discrete log-supermodular/log-submodular models in real-world settings, e.g., in a variety of computer vision tasks, continuous log-supermodular CRFs have not received much attention in practical applications.

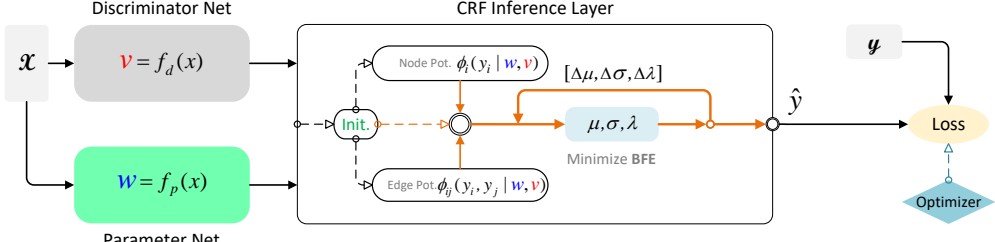

Figure 1: General architecture of the end-to-end NN+CRF model for structured prediction tasks.

# 3 Attractive Smoothing via Log-supermodular CRFs

The structured prediction problems considered here are supervised learning problems in which we are given data observations $x^{(1)}, \ldots, x^{(M)} \in \mathbb{R}^d$ and corresponding structured outputs $y^{(1)}, \ldots, y^{(M)} \in \mathcal{Y}$. The aim is to learn a model that can predict the structured outputs for novel data observations. In order to take advantage of the predictive power of deep neural networks in combination with the modeling power of CRFs for this task, we propose the simple end-to-end model shown in Figure 1. In this architecture, two neural networks, a discriminator NN and a parameter NN, are *trained to construct the parameters of a CRF* given a data observation $x$. Inference is then performed over the CRF in order to produce the final prediction, $\hat{y}$, which is then fed into a loss function. Specifically, the discriminator network produces a vector of outputs $v$ and the parameter network produces a vector of outputs $w$ and the CRF models the joint probability of the structured output $y$ given $v$ and $w$.

In applications, we will think of the discriminator net as a given DNN solution for the structured prediction task whose outputs we want to smooth and the parameter net will control the strength of the relationship between neighboring variables $y_i$ and $y_j$. As a motivating example, consider a task in which the input is a video and the aim is to determine whether or not each frame of the video contains an object from a known finite set, $F$. Consider a DNN that takes as input the $i^{th}$ frame of video and produces a vector $v^{(i)} \in \{0, 1\}^{|F|}$, where $v^{(i)} = 1$ indicates that the DNN predicts that object $f$ is in frame $i$. Applying this DNN for each frame of video generates a sequence of vectors and stacking the results produces a vector $v$, which we can treat as the output of the discriminator net in our architecture. If an object appears in frame $i$, it is likely to occur in frame $i + 1$ as well. We can construct a chain-structured CRF, with one binary random variable, $y_f^{(t)}$, for each $f \in F$ at each time slice $t$, whose potentials at time slice $t$ are given by

$$\phi_t^f(y_f^{(t)}|v^{(t)}, a) = \exp(a_f^{(t)}(y_f^{(t)} - v_f^{(t)})^2)$$
$$\phi_{t,t+1}^f(y_f^{(t)}, y_f^{(t+1)}|c) = \exp(c_f^{t,t+1} \cdot y_f^{(t)} \cdot y_f^{(t+1)}),$$

where $a$ and $c \geq 0$ are the outputs of the parameter net (indexed by $f \in F$ and time steps or pairs of time steps). The nonnegativity constraint on $c$ ensures that each potential function is log-supermodular. The effect of these potentials is to encourage the smoothness property over the sequence of video frames for the object detection task.

More generally, when applying attractive smoothing for structured objects $y \in S$ for some finite set $S \subset \mathbb{R}$, the univariate potential functions could be arbitrary (as univariate functions are always log-supermodular) and the pairwise potentials could be of the form $\exp(g(y_i, y_j|v, w))$, where $g(y_i, y_j|v, w)$ is a supermodular function of $y_i$ and $y_j$, i.e., $g(y_i, y_j|v, w) + g(y_i', y_j'|v, w) \leq g(y_i \wedge y_i', y_j \wedge y_j'|v, w) + g(y_i \vee y_i', y_j \vee y_j'|v, w)$ for all $y_i, y_j, y_i', y_j' \in S$. Note that in the most general case, we assume that the potential function values for each input/output pair are parameters of the CRF. The linear constraint on the pairwise potential functions could be added to the machine learning objective (discussed below) and enforced, with additional overhead, via projected gradient methods.

## 3.1 The Continuous Case

In the case of continuous models, similarities between random variables are often modeled using a multivariate Gaussian distribution. There are two primary reasons for this: (1) multivariate Gaussians represent a simple, well-understood parametric family and (2) Gaussian CRFs admit exact MAP

inference via matrix inversion. While this can be slow in practice for large models, in special sub-classes of Gaussian models, e.g., walk-summable Gaussian graphical models [33, 43], exact MAP inference can be done with Gaussian belief propagation, or similar approaches, without the expense of a full matrix inversion. Further, GPU versions of approximate message-passing algorithms, mean-field methods, or approximate variational inference methods, e.g., [55], can be used to make inference extremely practical even in large-scale models.

As an example of an approach based on walk-summable models, consider potentials of the form

$$\phi_i(y_i|v) \triangleq \exp(-(y_i - v_i)^2), \tag{1}$$

i.e., $v_i$ is the mean parameter of a Gaussian potential in the CRF, which is given by the discriminator net, and the parameterized pairwise potentials are of the form

$$\phi_{ij}(y_i, y_j|w) \triangleq \exp(-w_{ij} \cdot (y_i - y_j)^2). \tag{2}$$

The parameters, $w \geq 0$, are given as the output of the parameter NN. Note that the nonnegativity constraint on $w$ is sufficient to ensure that the joint distribution is a welk-summable Gaussian. This model is a generalization of the modeling approach taken by Liu et al. [31] and others.

The Gaussian CRF acts as a regularizer/smoother to encourage (or not) neighboring pixels to take similar values depending on how the parameter net selects the $w$ parameters. While the CRF in this example admits efficient inference, the modeling power of the potential functions is limited by the Gaussian requirement. Note that this model is log-supermodular by construction as the coefficients of the $y_i y_j$ terms is equal to $w_{ij}$, which is constrained to be nonnegative. If we relax the Gaussian requirement, the resulting log-quadratic model may not be integrable. However, in many practical applications, the structured outputs $y$ come from compact subset of $\mathbb{R}^n$ of the form $[a, b]^n$, and we can weaken the Gaussian requirement to only requiring log-supermodularity in such a setting. Specifically, consider potential functions of the form

$$\phi_i(y_i|v, a, b) = \exp(a_i \cdot y_i^2 + b_i \cdot y_i \cdot v_i) \tag{3}$$
$$\phi_{ij}(y_i, y_j|c) = \exp(c_{ij} \cdot y_i \cdot y_j), \tag{4}$$

where $v$ is the output of discriminator net and $w = (a, b, c)$ is the output of parameter net. From the second order condition for log-supermodularity, imposing the constraint that $c \geq 0$ is enough to ensure that each potential, and hence the entire model, is log-supermodular.

The log-supermodular model above contains the Gaussian model as a special case, but it need not produce a joint Gaussian model (it could correspond to a log-concave or log-convex distribution, but in general, it will be neither as the $a$'s can be either be positive or negative). We note that we will restrict log-supermodular models in the continuous case to log-quadratic functions for computational reasons as well as to avoid potential overfitting from much more flexible models. Despite this limitation, we are unaware of other works that take a similar approach in the continuous case.

## 4 End-to-end Training of Hybrid Models

Fitting the combined NN+CRF model to data requires selecting an appropriate loss function, e.g., negative conditional log-likelihood, squared error, etc., and performing gradient descent to minimize the loss. Different loss functions will result in different updates. Example loss functions that are popular in combined frameworks include the negative conditional log-likelihood (NLL), squared loss (SL), and the Huber loss (HL). No matter which loss function is selected, (approximate) inference will need to be performed on the CRF as part of the gradient update: For the NLL, this can be reduced to performing marginal inference, and for SL, HL, or similar discriminative losses, we need to differentiate the $\arg\max_y p(y|x)$ with respect to the NN parameters, which can be done by computing the maximum of $p(\cdot|x)$ using MAP inference and then applying the chain rule (construct partial derivatives with respect to the CRF parameters and then use backpropagation).

Approximate MAP inference in CRFs with continuous random variables can be performed using a variety of different approaches, e.g., mean-field variational inference, particle belief propagation [20, 27], or more general approximate variational inference strategies [17, 55]. Here we use the approximate variational methods, e.g., mean-field or the Bethe Free Energy (BFE), as they can be easily and efficiently implemented as a recurrent NN layer to seamlessly connect with the target

NN output. In both cases, we can think of the resulting (approximate) variational objective as an optimization problem over mean parameters, $\mu$, and covariances, $\Sigma$.

In the log-supermodular case, we could apply polynomial-time approximation schemes to estimate the MAP solution, e.g., [53, 1], but these strategies are likely to be too expensive, especially if we want to guarantee high accuracy. Practically speaking, the approximate variational methods are probably still preferred in practice, e.g., mean-field or the MAP inference approach of Xiong et al. [55], as these approaches are embarrassingly parallelizable and easy to implement as a *computation layer* in an ML framework like Tensorflow and Pytorch that can be seamlessly appended at the end of the original DNN (the overall computational overhead was around 10% in our experiments, see Table 4). The approximate variational methods are not guaranteed to return a global optimum, but the polynomial-time approximation schemes could be used to pick a good starting point for the approximate variational methods, e.g., by discretizing the domain and solving a simpler, approximate, discrete log-supermodular problem, in order to avoid poor local optima (we do not adopt this approach as it does not seem to be necessary to obtain performance improvements in practice).

## 5 Related Work

Historically, models based purely on CRFs have been used extensively for both stereo/monocular depth estimation, e.g., [24, 29, 32, 31, 45, 46, 49], as well as semantic image segmentation, e.g., [39, 38, 58]. The key challenge when applying CRFs for these problems is that fitting the CRF model and performing inference is typically intractable, which necessitates approximate inference, e.g., mean-field methods or particle belief propagation [20]. Prediction using the approximate methods can also be slow or inaccurate, e.g., particle BP may take a long time to converge (if it does at all). In addition, superpixels or subregions are often used to make the inference in these models scalable. Finally, features are typically hand-constructed, which often reduces practical performance. Motivated by the above limitations of pure CRF solutions, frameworks have sought to combine them with DNNs, e.g., for stereo matching [31, 24], for image segmentation [30, 61, 62, 9, 48, 28, 40] and for image colorization Messaoud et al. [37], or abandon CRFs altogether. While this list of hybrid approaches is not exhaustive for vision applications, it is illustrative of the limitations of many of the existing combined approaches. The end-to-end framework we will explore in the next section differs from existing DNN+CRF approaches in key ways: (1) we use a more general class of log-supermodular potentials, (2) our GPU implementation of the approximate inference routines makes our approach scalable even at the pixel level for vision tasks, (3) discretization is not used to reduce continuous inference to discrete inference, and (4) our models are trained with discriminative losses as opposed to minimizing the negative log-likelihood.

Similarly, there has been significant work on structured prediction that combines DNNs with CRFs. While our focus here is on log-supermodular CRFs, a number of general approaches have been proposed, and we highlight some of the most related here. Belanger et al. [4] consider a general framework for continuous energy minimization (SPEN). In contrast to the approach here which combines log-supermodular CRFs with DNNs, they consider a single black-box NN energy function (which can easily lead to overfitting in practice). Graber and Schwing [18] proposed an alternative to SPEN that allows problem structure to be incorporated to the learning process as part of a generalized structured SVM framework. The approach here is much simpler (and more targeted) and consequently inference is less time consuming (hence more scalable).

## 6 Experimental Evaluation

In this section, we evaluate the hybrid framework for attractive smoothing using existing DNN models developed for a variety of different machine learning applications. Below we describe the experimental setup in detail and then present qualitative and quantitative results for each of the application domains. All models were implemented using Tensorflow 2.2 and run on a $3\times$ NVIDIA Tesla V100 GPU. All source code, along with learned model weights, is available on GitHub[1]. In each application, we use the following skeleton.

**Discriminator Net:** a DNN that takes in the problem input and produces a structured output, $\overline{y}$.

---

[1] `https://github.com/motionlife/CRFBoostedDNN`

Table 1: Comparison of stereo matching models.

| Model | Sceneflow | | Virtual KITTI2 | |
|---|---|---|---|---|
| | > 1% | MAE | > 1% | MAE |
| GC-Net | 10.3% | 1.09 | 15.8% | 0.59 |
| GC-Net (SG) | 8.83% | 0.82 | 5.4% | 0.49 |
| GC-Net (LSM) | 7.92% | 0.77 | 3.84% | 0.31 |
| PSM-Net | 12.1% | 1.09 | 15.4% | 0.58 |
| PSM-Net (SG) | 9.00% | 0.78 | 9.3% | 0.50 |
| PSM-Net (LSM) | 6.77% | 0.67 | 5.69% | 0.38 |
| LEAStereo | 7.82% | 0.78 | 12.3% | 0.52 |
| LEAStereo (SG) | 7.75% | 0.75 | 9.2% | 0.47 |
| LEAStereo (LSM) | 7.68% | 0.75 | 7.54% | 0.50 |
| GA-Net-15 | 9.9% | .84 | - | - |

Table 2: Performance of DeepLabv3+ and its CRF variant on the coco-stuff validation data. Note that DeepLab as 41.4M parameters while the CRF adds an additional .5M parameters.

| Model | Pix. Acc. | MIoU |
|---|---|---|
| DeepLab-v3+ | 0.6334 | 0.2705 |
| DeepLab-v3+ (LSM) | 0.6553 | 0.2981 |

**Parameter Net:** a simple stacked hourglass convolutional net that takes the problem input and outputs the potential function parameters. See Table 3 in the appendix for details.

**CRF and Potentials:** Grid CRF of size $H \times W$, (which gives $HW$ nodes and $2HW$ edges), where $H$ and $W$ are the height and width of input image(s) for the DNN. We evaluate two different types of hybrid models: the typical Gaussian potentials from equations (1) and (2), which we denote as SG, and the log-supermodular potentials from (3) and (4), which we denote as LSM.

**Marginal inference:** Approximate inference is done via gradient ascent on the mean-field lower bound using the GPU scheme of Xiong et al. [55] with learning rate $\eta = 0.01$. The MAP assignment is computed from the joint distribution returned by the inference routine.

## 6.1 Stereo Matching

Given a pair of input images (left, right), the goal of stereo matching is to compute the disparity $d$ for each pixel in the reference image (the left image in this experiment). Given their speed and accuracy, end-to-end DNN solutions have become the most popular approach for stereo matching, e.g., GC-Net [23], PSM-Net [8], GA-Net [57], CSPNs [11], LEAStereo [13], CSPNs [12], HITNET [50], etc.

We selected several architectures for evaluation, GC-Net, PSM-Net, and LEAStereo as many recent architectures have been built from components of the first two and LEAStereo was near the top of the KITTI 2015 leaderboard at time of submission: GC-Net was the first architecture to learn a cost aggregation function (as opposed to hand-crafted cost aggregation functions) and also suggested using a differentiable soft-max while PSM-Net minimizes multiple losses during training (two are diverted from intermediate layers to prevent skewing of information).

**Model Details** The parameter net only takes the left image as input and outputs the potential parameters $w$ for SG and $a, b, c$ for LSM. The parameters $c$ and $w$ control how similar two neighbouring pixel's disparity should be. Intuitively, in SG, $w$, should be large for pixels in the same object surface and smaller for boundary pixels. All models are fit by minimizing the Huber loss:

$$L(\hat{d}, d) = \frac{1}{N} \sum_{n=1}^{N} H_\delta(\hat{d} - d), \text{ where } H_\delta(x) = \begin{cases} \frac{1}{2}x^2 & \text{for } |x| \leq \delta \\ \delta(|x| - \frac{1}{2}\delta) & \text{otherwise.} \end{cases} \tag{5}$$

We set $\delta = 1$ to match the setting of Chang and Chen [8]. For consistency, all models were trained from scratch with the Adam optimizer ($\beta_1 = 0.9, \beta_2 = 0.999$), learning rate set to 0.0001, and batch size 6. For each training batch, input images were randomly cropped to a dimension of $256 \times 512$. The maximum disparity ($D$) is set to 192.

**Datasets** We evaluate the proposed approach using the Scene Flow [34] and Virtual KITTI 2 [6], synthetic data sets are often used to pretrain DNNs that can be fine-tuned on much smaller real-world data sets such as KITTI [36, 35]. Detailed data set descriptions can be found in Appendix B. The training epochs for each data set are: 30 for Sceneflow and 25 for Virtual KITTI 2. Evaluations were done on full-sized images in the respective test sets.

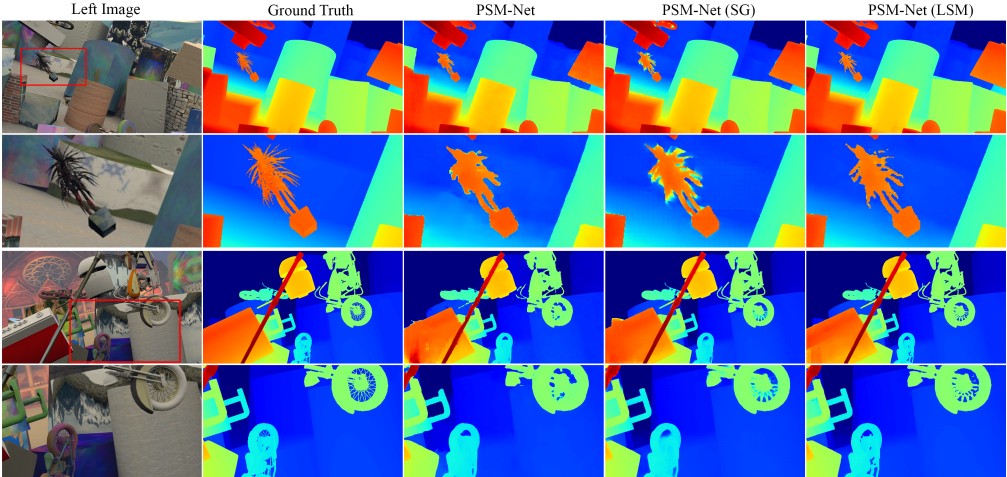

Figure 2: Visual comparison of PSM-Net and its two CRF hybrid models on Sceneflow test dataset.

**Results** For the experimental evaluation, we compare pure DNN architectures with combined DNN+CRF architectures on the data sets discussed above. We use the mean absolute error between the predicted disparity and the ground truth (MAE), sometimes referred to as end point error (EPE), and $k$-pixel threshold error percentage, i.e., the fraction of pixels whose absolute error is larger than $k$ pixels, as our evaluation metrics. The evaluation results are computed on all regions, both occluded and non-occluded areas, except if the pixel does not have a valid ground truth or is greater than a fixed maximum disparity setting ($D$). Our aim is to investigate whether or not the combined approach (1) results in significant performance improvements with respect to MAE and 1 pixel error over the pure NN approaches and (2) is efficient for both training and prediction.

The CRFs add an additional half-million or so parameters to the model but does not seem to result in a significant increase in prediction time despite the 10-20% increase in the number of parameters, e.g., GC-Net has 2.9M while GC-Net+CRF has 3.4M while the prediction time on the Sceneflow test set only increases by about .04s when the CRF is added. See Appendix C for additional speed/size details. We expect that similar observations would hold in other applications as long as the number of parameters of the parameter net and the CRF scale moderately with the dimension of $y$.

Table 1 presents the test set performance of the different models both with and without CRFs on the data sets discussed above. For comparison, we also include the published performance of GA-Net [57] on the Sceneflow data set (it was trained using the same loss function). We have the following observations. (1) Overall, the combined approaches outperform their respective pure approaches in terms of both MAE and 1 pixel error. On SceneFlow, the most significant performance improvements occurred for PSM-Net, both metrics were reduced between with LSM 40%-50%. And the resulting model had the best overall performance on Sceneflow. On Virtual KITTI2, adding LSM to GC-Net achieved the biggest performance improvements and also resulted in the best performing model on this data set. (2) Even the comparatively older architectures GC-Net and PSM-Net perform comparably or better than the more recent LEAStereo when a LSM CRF is used. PSM-Net (LSM) achieves the best Sceneflow performance while GC-Net (LSM) achieved the best Virtual KITTI2 performance. (3) The performance improvement obtained by hybrid approaches on Sceneflow applied to the LEAStereo model are minimal. We suspect that this is because the model search that was led to the construction of LEAStereo was tailored to this data set. However, on Virtual KITTI2, LEAStereo's base performance is only slightly better than GC-NET and PSM-Net, but the performance is improved significantly by adding a CRF. (4) The LSM CRF always outperforms the SG CRF, sometimes significantly. While SG is a special case of LSM, we observed that LSM performed better on both train and test data. This suggests that the added flexibility of LSM over SG does not result in significant overfitting. Additional quantitative results can be found in the supplementary material.

A qualitative comparison over the PSM-Net models can be found in Figure 2. As expected, the addition of the CRF appears to result in smoother local neighborhoods compared to the base model and better predictions on regions with fine details and thin areas, e.g., the spokes of wheels and the leaves of plants. Comparing SG with LSM, we see that SG tends to oversmooth, especially in regions

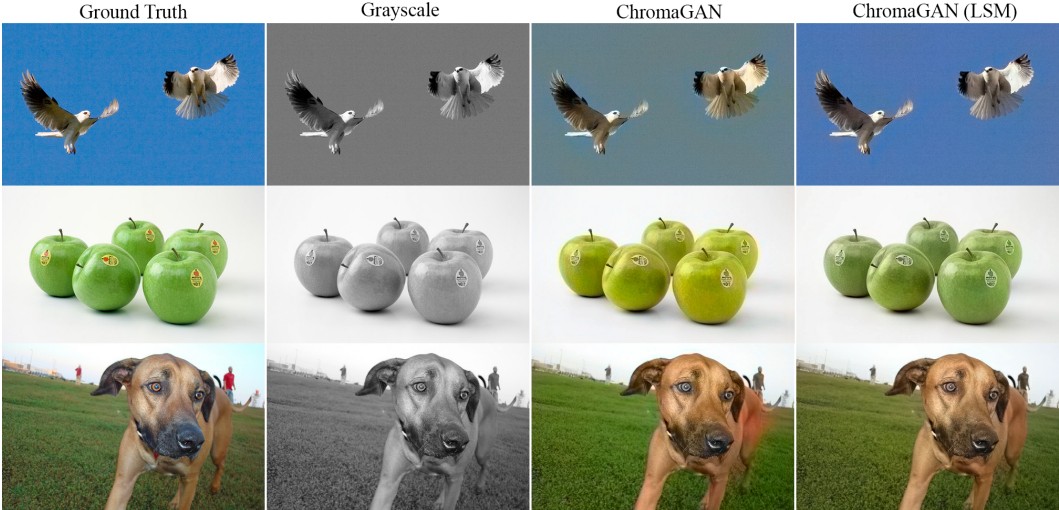

| Ground Truth | Grayscale | ChromaGAN | ChromaGAN (LSM) |

Figure 3: ChromaGAN versus ChromaGAN+CRF(LSM) on the ILSVRC2012 validation data set.

with a lot of fine details, as compared to LSM – this seems to be the primary difference between the two models. Additional qualitative results can be found in the supplementary materials.

## 6.2 Image Colorization

Next, we considered the problem of image colorization: given a grayscale image as input, the model needs to generate a full color image. Colorization is an ill-posed problem, requiring mapping a real-valued luminance image to a three-dimensional color-valued one, for which a unique solution may not exist. Many early DNN models used an encode-decoder, FCN architecture, [59, 3, 60], with some using classification loss, e.g., Zhang et al. [59], while others used smoothed $\ell_1$ loss, e.g., Zhang et al. [60] or $\ell_2$ loss/MSE [3]. More recent approaches have used generative adversarial networks (GANs) to generate colorful vivid images, e.g., [52]. One common problem with models trained to colorize images is that of color bleeding, see Figure 3 for an example of color bleeding produced by the ChromaGAN framework [52]. Our aim is to show that the hybrid approach can smooth local neighborhoods and mitigate color bleeding, at least in some cases.

**Model Details** To evaluate the hybrid approach, we used the ChromaGAN generative adversarial network for image colorization [52]. The ChromaGAN model consists of a pair of NNs: one is a generator network, which is responsible for generating the 2 missing color channels, the other is a discriminator network, which is used to judge how authentic the generated color images are. We added a grid-structured CRF whose parameters are selected via two input networks; a discriminator network that, in this case, is selected to be the generator network in ChrmoaGAN and a parameter net that takes the single channel image as input and produces potential parameters for the CRF. The output of the CRF inference, combined with input lightness channel, is sent to the discriminator during training. We report only the performance of LSM in this case as SG is a special case that only performed worse in the previous experiments.

**Dataset** ILSVRC2012 [44] contains roughly 1.3 million images categorized into 1K classes. The training/hyperparameter tuning follows that of Vitoria et al. [52], over epochs (10).

**Results** We evaluated the models using peak signal to noise ratio (PSNR) and structural similarity index measure (SSIM). The SSIM value was averaged across RGB channels when calculated against ground truth color images. Following Vitoria et al. [52], we evaluated the models on the first 1000 images from the ILSVRC2012 validation dataset, ChromaGAN has a 23.09 PSNR / 0.915 SSIM while Chroma-GAN (LSM) has a 24.19 PSNR / 0.927 SSIM (larger is better in both cases). Qualitatively, we found that the hybrid model can sometimes significantly reduce color bleeding. Figure 3 gives three examples of some visual improvements that were obtained by the hybrid model, which tends to

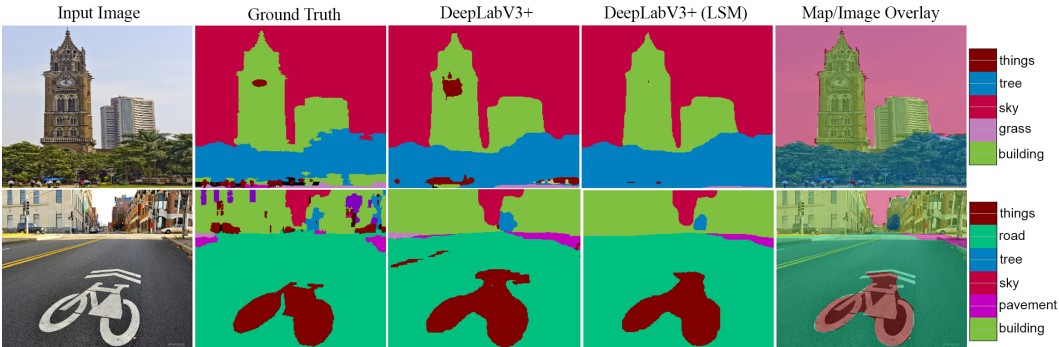

Figure 4: Segmentation results on the COCO-Stuff validation set. The CRF has a smoothing effect on the DNN output.

have less color bleeding around object boundaries, e.g., the orange spot near the dog's tail disappears after adding the CRF. Additional visual comparisons can be found in the supplementary material.

### 6.3 Semantic Segmentation

Finally, we consider the semantic segmentation problem, an image classification problem in which each pixel in the input image is assigned an individual category, e.g., road, tree, sky. The input is a single image, and the output is the per pixel classification.

**Model Details**   We used here is used the DeepLabv3+ [10] architecture to evaluate the hybrid approach. DeepLabv3+ is an encoder-decoder architecture, with an encoder network as a backbone to extract high level features and a decoder network to do up-sampling.The input image has size of $H \times W \times 3$, and the output logit tensor will be of size of $H \times W \times C$, where $C$ is the number of possible classes. Thus, for each pixel, there is a logit vector of size $C$ to represent the scores of that pixel belonging to the corresponding class. Given that each class score of neighbouring pixels should have dependency on each other, the previous DNN+CRF model that encourages similarities between neighbors should also make sense here. The only difference that is that is we apply a CRF for each class but enforce that those CRFs all share the same parameters.

**Dataset**   The COCO-Stuff dataet [7], which contains all images from the COCO-2017 dataset (118,287 training + 5K validation) with pixel-wise annotations for 91 stuff classes, e.g., sea, sky, river, etc., and 1 thing class for objects such as people, car, etc. COCO-Stuff does not distinguish between things. For training, $H = 512$, $W = 512$, $C = 93$ (91 stuff classes, 1 class unlabeled, 1 class for things). We use the Adam optimizer with learning rate 0.0001, both models were trained on this data set for 30 epochs with batch size 12.

**Results**   We evaluate the learned models using Mean Intersection-Over-Union (MIoU) and Pixel Accuracy. Table 2 displays the quantitative results: the LSM CRF yields a mild improvement with respect to both metrics. Figure 4 shows the qualitative results of this experiment: Adding the CRF smooths the segmentation map, eliminating some obvious prediction errors in the pure DNN model but over-smoothing in some cases. Although the CRF cannot correct more severe issues, e.g. distinct pavement and road, it does still produce a smoother overall result.

## 7   Discussion

While current trends seem to preference pure DNN solutions for challenging applications in computer vision and other domains, we have demonstrated that CRFs still provide a simple tool to smooth/regularize and encourage local consistency in DNN outputs. In addition, we showed that attractive smoothing via *continuous log-supermodular models* can yield significant performance improvements in practice. All of this is tied together in an end-to-end framework that scales using modern GPUs and approximate variational inference.

The flexible framework evaluated here can easily be incorporated into existing DNN approaches, and performance improvements in practice range from mild to significant with little additional overhead as part of the training or inference procedures on modern GPUs. Our hope is that these results provide enough empirical evidence to suggest that hybrid CRF+DNN frameworks be considered when designing new frameworks/architectures for which attractive smoothing is applicable. The experimental approach herein uses limited types of potential functions and limited types of smoothing for simplicity. It would be interesting to consider more expressive log-supermodular potential functions (including those of arity larger than two) in future work.

## Acknowledgments

This work was supported in part by the DARPA Perceptual Task Guidance (PTG) Program under contract number HR00112220005.

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
