# A   NN Structure

| No. | Layer Description | Output Shape |
|---|---|---|
| input | normalized left image | $H \times W \times 3$ |
| 1 | $3 \times 3$ conv | $H \times W \times 32$ |
| 2 | $3 \times 3$ conv, stride 2 | $H/2 \times W/2 \times 32$ |
| 3 | $3 \times 3$ conv | $H/2 \times W/2 \times 32$ |
| 4-5 | repeat 2-3 | $H/4 \times W/4 \times 48$ |
| 6-7 | repeat 4-5 | $H/8 \times W/8 \times 64$ |
| 8 | $3 \times 3$ deconv, stride 2 | $H/4 \times W/4 \times 48$ |
| 9 | $3 \times 3$ conv | $H/4 \times W/4 \times 48$ |
| 10-11 | repeat 8-9 | $H/2 \times W/2 \times 32$ |
| 12-19 | repeat 4-11 | $H/2 \times W/2 \times 32$ |
| residual connection | (3, 10), (5, 8), (7, 14), (9, 12), (11, 18), (13, 16) | |
| 20 | $3 \times 3$ deconv, stride 2 | $H \times W \times 32$ |
| output $w$ | $3 \times 3$ conv, no B.N., abs activaton | $H \times W \times 2(4)$ |

Table 3: Hourglass structured parameter net: layer configuration for stereo estimation. Each convolutional layer has batch normalization and ReLu activations, except for the output layer. Note, for LSM-CRF the dimension of output $w$ is 4.

# B   Stereo Matching Data Sets

**Sceneflow [34]** is a large synthetic data set which contains 35,454 training image pairs and 4,370 test pairs of $540 \times 960$ images. The data set contains images of random flying objects as well as synthetic driving images.

**Virtual KITTI 2 [6]** is an improved version of the original virtual KITTI synthetic driving data set of [15]. The data set contains 17,008 pairs of $375 \times 1242$ images from five different locations with variations based on time of day, weather conditions, and camera orientation. As Cabon et al. [6] did not divide the data into train and test splits, we used Scene01, Scene02, Scene18, Scene20 for training (14,848 image pairs) and Scene06 for testing (2,160 image pairs).

**DrivingStereo [56]** is a real-world driving scene data set consisting of $182,188$ labeled pairs of examples (174,437 pairs for training and 7,751 pairs for testing) and was the largest real-world data set available at time of submission. In contrast to the full resolution image results reported in their paper, the released data set has been down-sampled to half resolution, giving an average image size of $400 \times 881$ pixels. In addition to the sequential training data, the authors also hand-selected 2000 frames with 4 different weather conditions (sunny, cloudy, foggy, rainy). We consider training using these two different training sets and evaluation is performed on the entire test set.

**KITTI 2012 and 2015 [16, 36, 35]** are both real-world driving data sets containing around 200 training image pairs and 200 test image pairs each with an average image size of $375 \times 1242$. Sparse ground truth is obtained by LiDAR scanning points. Given its small size, most models fit to these two data sets are first pretrained on a larger data set before being fine-tuned on KITTI.

# C   Stereo Matching Model Sizes

| Model | #Params. | Time(s) |
|---|---|---|
| GC-Net | 2.9M | 0.25 |
| GC-Net + CRF | 3.4M | 0.29 |
| PSM-Net | 5.2M | 0.31 |
| PSM-Net + CRF | 5.7M | 0.35 |

Table 4: Number of model parameters and prediction time on the Sceneflow test set for the four different architectures. All evaluated on a single NVIDIA Tesla V100 GPU.

# D   KITTI 2012 and 2015 Results

Additional results on KITTI 2012 and 2015.

A qualitative comparison of the four different architectures on test set images from Sceneflow, Virtual KITTI 2, and DrivingStereo can be found in Figure 5. In addition to producing smoother results, we observe that the combined DNN+CRF architectures typically outperform pure DNN architectures in subregions that have reflective surfaces or thin structures. To further investigate this, we compared the performance of the architectures after being pretrained on Sceneflow and then fine-tuned on the KITTI 2012 data set. The results of this evaluation can be found in Table 5 of the supplementary material. The performance gain achieved by adding the CRF is significant for both GC-Net and PSM-Net, which provides strong quantitative evidence in support of the qualitative observations in Figure 5. At time of submission, the PSM+CRF performance on reflective regions (4-px error threshold) would place it at rank #7 on the KITTI 2012 benchmark leader board, a significant jump from PSM-Net's rank of #84 at time of submission. A visual comparison of the PSM+CRF model versus GA-Net-15 on KITTI 2012 and 2015 can be found in Figures 6 and 7 respectively.

| Model | Out-Noc(3px) | Avg-Noc | Avg-All |
|---|---|---|---|
| GC | 10.8% | 1.8px | 2.0px |
| GC+CRF | 7.65% | 1.2px | 1.4px |
| PSM | 8.36% | 1.4px | 1.6px |
| PSM+CRF | **5.63%** | **1.1px** | **1.2px** |
| GA-Net-15 | 7.87% | 1.3px | 1.5px |

Table 5: Evaluation results on the KITTI 2012 benchmark over reflective regions. The CRF appears to give a significant performance boost in these regions.

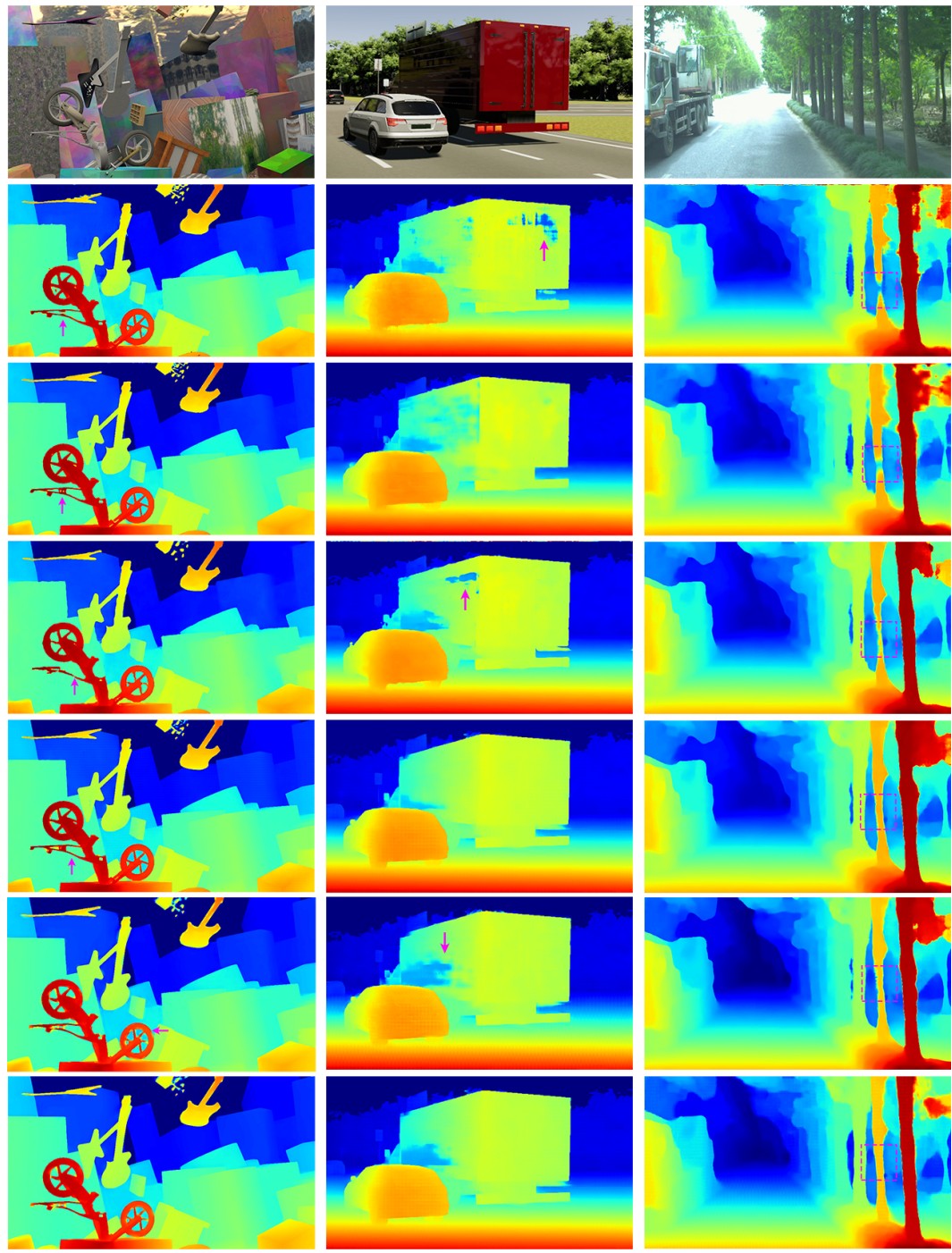

Figure 5: **Qualitative results for Sceneflow, Virtual KITTI 2 and DrivingStereo\***. From top to bottom: left input images, GC-Net, GC+CRF, PSM-Net, PSM+CRF, LEAStereo, LEAStereo+CRF. The CRF yields apparent improvement on thin structure and reflective regions.

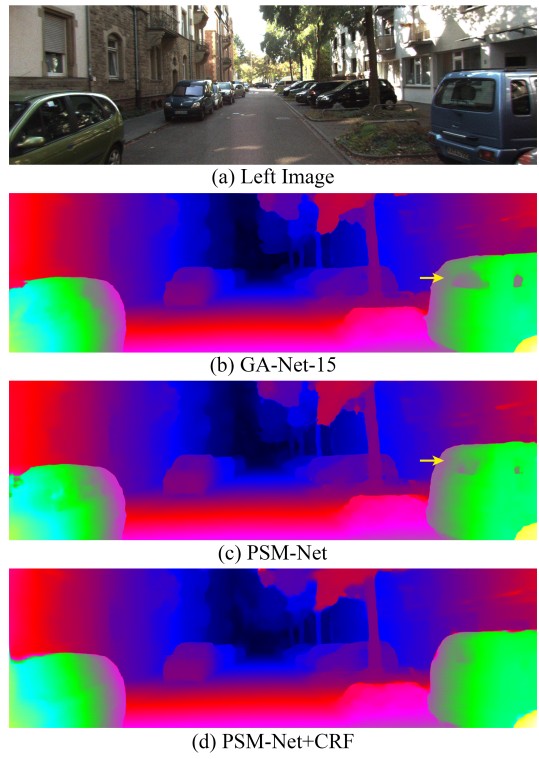

(a) Left Image

(b) GA-Net-15

(c) PSM-Net

(d) PSM-Net+CRF

Figure 6: Comparisons of different architectures on reflective regions on an image from the KITTI 2012 test set.

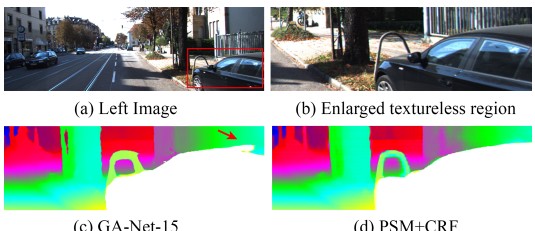

(a) Left Image        (b) Enlarged textureless region

(c) GA-Net-15        (d) PSM+CRF

Figure 7: Comparisons of GA-Net-15 with PSM+CRF on large textureless regions on an image from the KITTI 2015 test set.