# OpenReview forum: "Boosting the Performance of Generic Deep Neural Network Frameworks with Log-supermodular CRFs"
_NeurIPS.cc/2022/Conference — NeurIPS 2022 Accept_

### Official Review · Reviewer_vmzk · 2022-07-07

**Rating:** 5
**Confidence:** 2
**Soundness:** 3 good
**Presentation:** 2 fair
**Contribution:** 2 fair

**Summary:**

The authors present log-supermodular CRF as a smoothing layer for a wide class of applications, including optical flow estimation, activity recognition, colorization, and segmentation. They show in experiments that the proposed module can improve the performance of the model over all the mentioned tasks.

**Questions:**

- Are the potential functions you presented in Equations 1-4 log-supermodular? Why are they would be helpful for smoothing?
- Could you briefly explain what "attractive" means in this context?
- In Figure 1, what is $\mu,\sigma, \lambda$ and  $\Delta\mu,\Delta\sigma, \Delta\lambda$? you may need to give an explicit form of BFE in your methodology.
- Why not compare with some deep network with ad-hoc smoothing techniques?
- In Figure 2, I can only tell the first and second rows have distance differences, while the rest rows do not show much difference.
- In Figure 3, I cannot tell which colorization is better. Are there any other examples that can show the impact of the proposed method?
- In Figure 4, smooth for segmentation may remove all the detailed small blocks. It may be helpful but not necessary.

**Ethics Review Area:**

["I don’t know"]

**Limitations:**

The author may mention if their attractive smoothing idea can be (not-)helpful for other CRF applications in text, like part-of-speech tagging and dependency parsing.

**Strengths And Weaknesses:**

Strengths:
- easy to read through and solid experiments.

Weakness:
- the authors only compare the deep models with and without attractive smoothing. I would encourage them to consider some other adaptive, ad-hoc smoothing methods as baselines.
- It's hard to tell the improvement through the figures for tasks like colorization and segmentation. Also, it is questionable whether segmentation needs smoothing.

---

> ### Author Response · Authors · 2022-07-31
> **Attractive models and qualitative observations**
>
> Thanks for your suggestions!
>
> >> Could you briefly explain what "attractive" means in this context?
> >> Are the potential functions you presented in Equations 1-4 log-supermodular? Why are they would be helpful for smoothing?
>
> Log-supermodular graphical models are sometimes referred to as attractive graphical models.  The terminology is motivated by the observation that the potential functions in these models encourage/prefer connected variables to have the same values.  See for example,
>
> E. B. Sudderth, M. J. Wainwright, and A. S. Willsky, “Loop series and Bethe variational bounds in attractive graphical models,” in NIPS, 2007.
>
> Yes, equations (1)-(4) are log-supermodular (see the in-text discussion).  They encourage neighboring variables in the CRF to take similar values.  How much they encourage similarity depends on the magnitude of the parameters.  The resulting inference task then aims to find an assignment to the variables that is most likely given their local preferences.
>
> >> In Figure 1, what is ...? you may need to give an explicit form of BFE in your methodology.
>
> We use the same notation as [55]:  mu, sigma, and lambda are the parameters of the variational approximation.  There is some discussion in the text, e.g., line 164, but due to space constraints, we refer you to [55] for the complete details.
>
> >> Why not compare with some deep network with ad-hoc smoothing techniques?
>
> There are different kinds of smoothing that are used in the literature.  Our approach is generic in the sense that the DNN is treated like a black box.  If you wanted to perform some type of DNN ad-hoc smoothing, you could still stick a CRF a the end of the whole process.  In general, the smoothing that is accomplished by the CRF model depends on its structure and what knowledge that you are trying to encode.  This is likely to produce different results than a general ad-hoc smoothing approach that is application independent.  Though, it still may be interesting to compare.
>
> >> In Figure 2, I can only tell the first and second rows have distance differences, while the rest rows do not show much difference.
>
> As all of the models have relatively high accuracy for this task, it can be difficult to see the differences at first glance (some zooming may be required).  The second and last rows are a blow-up of the first and third that shows the difference on thin regions (in this case, the leaves of a plant or the spokes on a bike wheel).
>
> >> In Figure 3, I cannot tell which colorization is better. Are there any other examples that can show the impact of the proposed method?
>
> Goodness of colorization is somewhat subjective.  However, one of the common issues with color bleeding, i.e., two distinct objects or regions share colors in the prediction when they don't in the ground truth.  For example, in the last row, the color of the back of the dog blends into the background under ChormaGAN.  Again, some zooming may be required to see the differences between the two outputs.
>
> >> In Figure 4, smooth for segmentation may remove all the detailed small blocks. It may be helpful but not necessary.
>
> We agree -- though it could be somewhat application dependent.  We chose this application simply to illustrate the effect of the proposed approach.

---

### Official Review · Reviewer_xqVe · 2022-07-08

**Rating:** 6
**Confidence:** 1
**Soundness:** 3 good
**Presentation:** 3 good
**Contribution:** 3 good

**Summary:**

The paper proposes the use of super-modular CRF to regularize a DNN model for applications where smoothing is important. The intuition is that while DNNs are easier to scale, they do not take into account the natural constraints in the task e.g. two neighboring image pixels need to have similar colors. In this DNN-CRF hybrid model, there are two DNNs for generating the feature vectors and the CRF parameters respectively, and a CRF-based inference layer. Results are reports on three tasks: stereo matching, image colorization and semantic segmentation. On stereo matching, adding the CRF regularizer improves the performance of three DNN baselines. On semantic segmentation, CRF gives a small boost to the DNN. On image colorization, adding the CRF improves the peak signal to noise ratio.


**Questions:**

* It would be useful to provide more details on the DNN-CRF hybrid training including the algorithm, the variational method, update steps, any practical tricks to ensure convergence, etc.
* What is the impact on training time due to the CRF regularizer?
* Would it be possible to report a metric that indicates whether the proposed method has improved the smoothness objective on each of the tasks?



**Limitations:**

There is no explicit discussion of limitations in the paper. Hence, it would be useful to add a paragraph on limitations to the conclusions section. The authors should also add a sentence about the potential negative societal impact in the conclusions.


**Strengths And Weaknesses:**

Strengths:
* Novel use of supermodular CRFs to improve DNN performance across tasks where smoothing is important
* Performance improvements on three tasks

Weaknesses:
* It would be useful to compare performance against alternative neural approaches e.g. RNN or transformer, that can take into account correlations across time steps.
* The paper does not report the impact on training time due to the CRF regularizer.
* The paper states that "Our approach applies in both discrete and continuous settings" but the approach is evaluated only for scenarios with continuous inputs.
* It would be useful to provide more details on the hybrid model training e.g. are there practical tips to ensure convergence? Does the DNN/CRF training need to be done in some alternating fashion etc?
* The appendix is not included in the paper.

---

> ### Author Response · Authors · 2022-07-31
> **A brief response**
>
> Thanks for your feedback!
>
> >> It would be useful to compare performance against alternative neural approaches e.g. RNN or transformer, that can take into account correlations across time steps.
>
> Thanks for the suggestion.  In the selected applications, we were only looking at single frame predictions.  If we instead were looking at, say, depth estimation in video, the competitors would likely be RNN or transformer models.  In such cases, we would need to use a CRF that can capture dependence over time.  We leave these applications for future work.
>
> >>It would be useful to provide more details on the DNN-CRF hybrid training including the algorithm, the variational method, update steps, any practical tricks to ensure convergence, etc.
>
> Due to space considerations, we will refer you to [55] for the details of the variational approach.  The algorithmic approach first computes the output of the CRF for a given input and then performs backpropagation in the standard way.  No special tricks outside of those typical for training with backpropagation, e.g., adaptive step sizes, etc., were used to ensure convergence.  The key idea in [55] is simply to encode each iteration of the variational scheme as an additional computation layer in Tensorflow.
>
> >>What is the impact on training time due to the CRF regularizer?
>
> We observed that the effect of adding the CRF is minimal in our experiments.  The biggest bottleneck is that, in order to do backpropagation, the prediction of the CRF needs to be computed in addition to the output of the DNN.  This turns out not to add much overhead compared to the pure DNN framework.  For example, in the stereo estimation task, in GC-Net with 2.9M parameters prediction takes approximately 0.25 seconds on average and in GC-Net + CRF with 3.4M parameters prediction takes about 0.29 seconds on average.  In general, we observed that the computational overhead of the CRF is about 10% above the pure DNN solution.
>
> >>Would it be possible to report a metric that indicates whether the proposed method has improved the smoothness objective on each of the tasks?
>
> This is a good question.  Right now we are only assessing smoothness qualitatively as we used the same evaluation metrics that were used for the DNN architecture in each application.  The appropriate notion of smoothness may be application dependent.

---

> > ### Comment · Reviewer_xqVe · 2022-08-09
> > **Responses ok but comparisons weak**
> >
> > Thanks to the authors for their responses. I am satisfied with some of them but I still have concerns:
> > 1) It is not entirely clear if the baselines in this paper are strong enough. I'm not an expert in this field but it seems there should
> > be existing (potentially imperfect) alternatives to perform smoothing with DNN that would serve as reasonable baselines.
> > As an example, this paper (https://link.springer.com/content/pdf/10.1007/s11042-020-10468-6.pdf) uses Total Variational Loss to prevent excessive contrast in the image using a DNN model for medical image colorization.
> > 2) As I mentioned in my previous comments. The paper doesn't have explicit metrics for smoothness.
> >
> > Therefore, I am keeping my original scores as is.

---

### Official Review · Reviewer_WF55 · 2022-07-21

**Rating:** 6
**Confidence:** 2
**Soundness:** 3 good
**Presentation:** 3 good
**Contribution:** 3 good

**Summary:**

Paper proposes an attractive smoothing algorithm via log-supermodular CRFs, which takes the CRF as a regularizer for the DNNs. . Extensive experiments on a range of applications demonstrate that even conventional models outperform more recent ones with the addition of an attractive CRF.

The paper is written clearly, both the quantitative and qualitative analysis experiments are well done, which can fully verify the effective of the attractive smoothing algorithm. However, I'm not an expert in this field, so I'm not able to appraise it from the aspects of innovation, algorithm design, and whether it has a positive role in promoting the research field.

**Questions:**

N/A

**Strengths And Weaknesses:**

N/A

---

### Meta-Review · Area_Chair_gxYR · 2022-08-30

**Recommendation:** Accept
**Confidence:** Less certain

**Metareview:**

The paper proposes to use   Log-supermodular CRFs  to smooth the DNN models. The paper is well motivated and conduct extensive experiments to  verify the effective of the attractive smoothing algorithm.
It could be better if the paper conduct more experiments based on different networks such as RNN and Transformer.  Besides, this paper only explores the proposed attractive smoothing. It should make a comparision with other smoothing methods.


**Award:**

No

---

### Decision · Program_Chairs · 2022-09-14

Accept